# A Comparison of Three Machine Learning Methods for Multivariate Genomic Prediction Using the Sparse Kernels Method (SKM) Library

**DOI:** 10.3390/genes13081494

**Published:** 2022-08-21

**Authors:** Osval A. Montesinos-López, Abelardo Montesinos-López, Bernabe Cano-Paez, Carlos Moisés Hernández-Suárez, Pedro C. Santana-Mancilla, José Crossa

**Affiliations:** 1Facultad de Telemática, Universidad de Colima, Colima 28040, Mexico; 2Centro Universitario de Ciencias Exactas e Ingenierías (CUCEI), Universidad de Guadalajara, Guadalajara 44100, Mexico; 3Facultad de Ciencias, Universidad Nacional Autónoma de México (UNAM), México City 04510, Mexico; 4Instituto de Ciencias Tecnología e Innovación, Universidad Francisco Gavidia, El Progreso St., No. 2748, Colonia Flor Blanca, San Salvador CP 1101, El Salvador; 5International Maize and Wheat Improvement Center (CIMMYT), Texcoco 56237, Mexico; 6Colegio de Postgraduados, Montecillo 56230, Mexico

**Keywords:** multi-trait, statistical machine learning, genomic selection, plant breeding, multi-environment

## Abstract

Genomic selection (GS) changed the way plant breeders select genotypes. GS takes advantage of phenotypic and genotypic information to training a statistical machine learning model, which is used to predict phenotypic (or breeding) values of new lines for which only genotypic information is available. Therefore, many statistical machine learning methods have been proposed for this task. Multi-trait (MT) genomic prediction models take advantage of correlated traits to improve prediction accuracy. Therefore, some multivariate statistical machine learning methods are popular for GS. In this paper, we compare the prediction performance of three MT methods: the MT genomic best linear unbiased predictor (GBLUP), the MT partial least squares (PLS) and the multi-trait random forest (RF) methods. Benchmarking was performed with six real datasets. We found that the three investigated methods produce similar results, but under predictors with genotype (G) and environment (E), that is, E + G, the MT GBLUP achieved superior performance, whereas under predictors E + G + genotype × environment (GE) and G + GE, random forest achieved the best results. We also found that the best predictions were achieved under the predictors E + G and E + G + GE. Here, we also provide the R code for the implementation of these three statistical machine learning methods in the sparse kernel method (SKM) library, which offers not only options for single-trait prediction with various statistical machine learning methods but also some options for MT predictions that can help to capture improved complex patterns in datasets that are common in genomic selection.

## 1. Introduction

Genomic selection (GS) is a predictive methodology that is revolutionizing plant breeding for the selection of candidate individuals using reference information (training sample) that contains phenotypic and genotypic information with which a statistical machine learning model is trained. Then, with this trained model, predictions for candidate individuals that were not phenotyped are obtained using genotypic data. The empirical evidence of the power of GS methodology to accelerate the rate of genetic gain with respect to traditional breeding [1,2,3] continues to grow in crops such as maize, wheat, chickpea, etc. [4,5,6,7]. The boom of GS has been catalyzed by a significant reduction in the cost of genotyping technologies, such as genome-wide markers. Furthermore, because GS methodology has been shown to be particularly useful where traits need to be phenotyped, it is used not only for many annual crops but also in some long-lived species.

However, there are still many factors that need to be carefully considered for the successful practical application of GS. These factors include (1) the size and degree of relatedness of the training and testing sets, (2) the complexity of the trait that needs to be predicted, (3) the quality of the phenotypic and marker data available for the training process, (4) the goal of prediction (for example, predicting new lines in a complete year or location or predicting some lines that are missing in some environments but present in others) and (5) the statistical machine learning method that needs to be implemented [2,3,8].

Statistical machine learning methods include statistical learning and machine learning methods; many such methods (and models) have been proposed, compared, and implemented [4]. Some of the first models used were linear mixed models; then, their Bayesian counterparts and conventional Ridge regression methods were introduced. Many algorithms have been implemented to date for genomic selection (random forest (RF), gradient boosting machine (GBM), deep learning (DL), partial least squares (PLS) etc.). Many of these methods have been achieved success in genomic prediction. For example, mixed models are still the predominant statistical machine learning tools for genomic prediction in animal science. Due to their ability to incorporate prior information, Bayesian methods are powerful prediction tools in plant and animal science. Bayesian methods have the advantage that with default hyperparameters, they work well in many circumstances and are very robust in the face of prediction problems, even with small datasets, and can be used on a personal computer without the need for sophisticated servers with many cores. Additionally, conventional mixed models and Bayesian methods (GBLUP, BayesA, BayesB, BayesC and Bayesian Lasso) have recently been implemented successfully using kernels, which increase the power of these methods, as with the use of non-linear kernels, they are able to capture not only linear but also non-linear patterns more efficiently [5,8].

With respect to the RF, GBM and DL methods, they are currently popular in the machine learning community and are also used for genomic prediction with promising results. The power of these methods is that they can naturally capture non-linear patterns and, in the case of RF and GBM, produce competitive predictions, even with default hyperparameters. However, DL methods generally require large datasets and a complex tuning process to obtain good or above state-of-the-art prediction performance. However, with enough effort in the tuning process, these statistical machine learning methods produce competitive predictions. Therefore, for successful implementations of these machine learning methods, it is of paramount importance to dedicate a considerable amount of time to the tuning process. About the PLS method, few applications have been reported in the context of genomic prediction; however, this method has a long history of prediction in the chemometrics and biological sciences due to its ability to efficiently deal with datasets with correlated inputs and more independent variables than observations. This method requires tuning of the number of principal components to be used; however, because this is the only hyperparameter to tune, the selection of this optimal hyperparameter does not require expensive computation [8].

However, empirical evidence shows that when more than one trait of interest needs to be predicted, multi-trait (MT) models are more efficient than single-trait (ST) analysis [9]. Some of the reasons as to why MT models are preferred over ST models include that (1) they capture complex relationships between correlated traits more efficiently; (2) they take advantage of the degree of correlation between lines and traits; (3) MT models offer better interpretability than ST models; (4) they are computationally more parsimonious to train than ST models; (5) more precise estimates of random effects of lines and genetic correlations between traits are obtained, which allows for improvement of the index selection; (6) they become more efficient for indirect selection as the precision of genetic correlation parameter estimates increases; and (7) they improve hypothesis testing because they reduce type I and II errors. A type I error (false positive) denotes the rejection of a null hypothesis that is true; a type II error (false negative), on the other hand, denotes the failure to reject a null hypothesis that is false [10,11].

As mentioned above, MT models, in addition to helping increase prediction accuracy, also increase statistical power by improving the accuracy of parameter estimates [12,13,14] and are useful for the prediction of primary traits (difficult or expensive to phenotype) using available secondary traits, especially when they have low heritability. However, the MT best linear unbiased predictor model was originally proposed by Henderson et al. [12] to reduce selection bias.

Despite the advantages of MT models, ST models are more popular for genomic prediction for some of the following reasons: (a) lack of efficient software for MT genomic prediction; (b) limitations in computational resources, as MT models require more resources than ST models; (c) lack of MT models to efficiently model genotype × environment interactions (GE), as traits have different response patterns depending on the environment; (d) MT make more assumptions than ST models that are not easy to achieve; (e) MT models have more issues of convergence than ST models; and (f) in general, the implementation of MT models is more challenging due to the complexity of the underlying datasets. Therefore, the successful implementation of MT models requires more effort than ST models [10,11].

For the above reasons, the use of MT models continues to increase for genomic prediction. MT mixed models and Bayesian models are the most popular methods used for genomic prediction. However, due to the advantages offered by these MT models, other alternative multivariate statistical machine learning methods have been explored. For example, Montesinos-López et al. [15,16] explored the use of deep learning methods for multi-trait genomic prediction. Additionally, due to its power in modeling complex biological data with a high degree of collinearity [17], the MT-PLS method has been explored for genomic prediction with promising results.

Based abovementioned considerations, the objectives of this paper are twofold. The first goal is to compare the prediction performance of three multi-trait machine learning methods; one method is popular for genomic prediction [the Bayesian multi-trait (MT) genomic best linear unbiased predictor (GBLUP)], the other method is employed in the chemometrics and omics area (the MT partial least squares (PLS) method), and the last method is used in in all the statistical machine learning fields [random forest (RF)]. The second goal is to illustrate to scientists in plant breeding, animal science and related areas the power of the sparse kernel method (SKM) library [18] with respect to efficient implementation of multi-trait prediction models.

## 2. Materials and Methods

### 2.1. Models

#### 2.1.1. Bayesian MT-GBLUP Model

This model is given by
$$Y=1_{n}\mu^{T}+X_{E}\beta_{E}+Z_{L}g+Z_{EL}gE+\epsilon \;\left(1\right)$$
where Y is the matrix of phenotypic response variables of order $n\times n_{T}$ and ordered first by environments and then by lines; $n_{T}$ denotes the number of traits; $1_{n}$ is a matrix of ones of order $n\times n_{T}$, where $n_{T}$ denotes the number of traits; $\mu^{T}$ is a vector of intercepts for each trait of length $n_{T}$, T denotes the transpose of a vector or matrix, that is, $\mu ={\left[\mu_{1},\;\ldots ,\;\mu_{n_{T}}\right]}^{T};\;\;X_{E}$ is the design matrix of environments of order n×I; I denotes the number of environments; $\beta_{E}$ is the matrix of beta coefficients for environments with a dimension of $I\times n_{T}$; $Z_{L}$ is the design matrix of lines of order n×J; J denotes the number of lines; g is the matrix of random effects of lines of order $J\times n_{T}$ distributed as $g∼MN_{J\times n_{T}}\left(0,G,\Sigma_{T}\right)$, that is, with a matrix-variate normal distribution with parameters M=0, U=G and $V=\Sigma_{T}$; G is the genomic relationship matrix [19] built with marker data of order J×J; $\Sigma_{T}$ is the variance–covariance matrix of traits of order $n_{T}\times n_{T}$; $Z_{EL}$ is the design matrix of the genotype × environment interaction of order n×JI; gE is the matrix of the genotype × environment interaction random effects distributed as $gE∼MN_{JI\times n_{T}}\left(0,G\;⊗\;\Sigma_{E},\Sigma_{T}\right)$, where $\Sigma_{E}$ is a diagonal variance-covariance–matrix of environments of order I×I, and $G\;⊗\;\Sigma_{E}$ is the Kronecker product of the lth type of kernel matrix of lines and the environmental relationship matrix; and ϵ is the residual matrix of dimension$\;n\times n_{T}$ distributed as $\epsilon ∼MN_{n\times n_{T}}\left(0,I_{IJ},R\right)$, where R is the residual variance–covariance matrix of order $n_{T}\times n_{T}$.

#### 2.1.2. Random Forest (RF) Model

RF is an alteration of bootstrap aggregation that builds a large collection of trees and averages out the results. Each tree is built using a splitting criterion (loss function), which should be appropriate for each type of response variable (continuous, binary, categorical and count). To train data [20], RF takes B bootstrap samples and randomly selects subsets of independent variables as candidate predictors to split tree nodes. Each decision tree minimizes the average loss function in the bootstrapped (resampled) data and is built-up using the following algorithm:

For b=1,…,B bootstrap samples $\left\{y_{b},X_{b}\right\}$

**Step 1.** From the training dataset, draw bootstrap samples of size $N_{train}$.

**Step 2**. With the bootstrapped data, grow a random forest tree ($T_{b}$) with the specific splitting criterion (appropriate for each response variable) by recursively repeating the following steps for each terminal node of the tree until the minimum node size (minimum size of terminal nodes) is reached.

Randomly draw mtry out of the m independent variables (IVs); mtry is a user-specified parameter and should be less than or equal to p (total number of IVs);Select the best independent variable among the mtry IVs.Split the node into two child nodes. The split ends when a stopping criterion is reached, for instance, when a node has less than a predetermined number of observations. No pruning is performed.

**Step 3**. The ensemble of trees is obtained as ${\left\{T_{b}\right\}}_{1}^{B}$.

The predicted value of testing set (${\hat{y}}_{i}$) individuals with input $x_{i}$ is calculated as ${\hat{y}}_{i}=\frac{1}{B}\overset{B}{\underset{b=1}{\sum }}T_{b}\left(x_{i}\right)$ because our multi-trait response variable is continuous. Readers are referred to Breiman [20] and Waldmann [21] for details of the theory of RF. This implementation used the minimization of the multivariate sum of squares at the splitting criteria, which is an appropriate approach for multi-trait response variables.

#### 2.1.3. Multi-Trait Partial Least Square (MT-PLS) Method

PLS is a multi-trait regression statistical technique introduced by Wold [22] in the fields of econometrics and chemometrics. PLS is efficient for dealing with the p>n problem, i.e., when the number of observations (n) is considerably less than the number of explanatory variables (p), which are often highly correlated. The multi-trait version of PLS is suitable for relating a matrix of response variables (Y) to a set of explanatory variables (X) [23].

In PLS, regression analysis is performed by regressing Y on X but by regressing Y on T, where T is the latent variables (LVs), latent vectors or X scores; however, these LVs are obtained iteratively. The basic steps to compute the LVs are given next:

**Step 1**. Initialize two matrices, E = X and F = Y. Center and normalize each column of E0 and F0.

**Step 2**. Form a cross-product matrix $\left(S=X^{T}Y\right)$ and determine its singular value decomposition (SVD). The first left and right singular vectors, w and q, are used as weight vectors for X and Y, respectively, to obtain scores t and u:(1)t=Xw=Ew
(2)u=Yq=Fq
where E and F are initialized as X and Y, respectively. The X scores (t) are often normalized:(3)$$t=t/\sqrt{t^{T}t}$$

The Y scores (u) are not necessary in the regression but are often maintained for interpretation purposes.

**Step 3.** Next, X and Y loadings are obtained by regressing against the same vector (t):(4)$$p=E^{T}t$$
(5)$$q=F^{T}t$$

**Step 4.** Having extracted the first latent vector and the corresponding loading vectors, the matrices E and F are deflated by subtracting information related to the first latent vector. This produces deflated matrices $E_{n+1}$ and $F_{n+1}$, as shown in the calculations below.
(6)$$E_{n+1}=E_{n}-tp^{T}$$
(7)$$F_{n+1}=F_{n}-tq^{T}$$

**Step 5**. Calculate the cross-product matrix of $E_{n+1}$ and $F_{n+1}$ as in Step 2. With this new cross-product matrix, repeat steps 3 and 4 and save the resulting w, t, p and q vectors to form the next columns of matrices: **W**, **T**, **P** and **Q**, respectively. This yields the next component. Then, repeat the above steps until the deflated matrices are empty or the necessary number of components have been extracted. Then the algorithm stops.

Note that the columns of matrix W cannot be compared directly; they are derived from successively deflated matrices (E and F) as demonstrated in the previous five steps. Therefore, after obtaining all the columns of W, R is computed as:(8)$$R=W{\left(P^{T}W\right)}^{-1}$$

Finally, using R, the latent variables related to the original X matrix can be computed as:(9)T=XR

Next, because Y was regressed on T, the resulting beta coefficients are $b={\left(T^{T}T\right)}^{-1}T^{T}Y$. However, to convert these back to the realm of the original variables (X) matrix R is pre-multiplied by the beta coefficients (b) because T=XR:(10)B=R b

To improve the performance of the PLS method, only the first a components are used. Because regression and dimension reduction are performed simultaneously, B, T, W, P and Q are all part of the output. Both X and Y are taken into account when calculating the LVs in T. Moreover, they are defined so that the covariance between the LVs and the matrix of response variables is maximized. Finally, predictions for new data $\left(X_{new}\right)$ should be calculated according to:(11)$${\hat{Y}}_{new}=X_{new}B=X_{new}Rb=T_{new}b$$
where $T_{new}=X_{new}R$. Usually, the optimal number of components needs to be determined by cross validation. We used the root means squared error of prediction (RMSEP), which was minimized with 10-fold cross validation in the training dataset and for each value of LV [24].

The input used under multi-trait PLS is the concatenation of the following three augmented matrices: $X_{L}L_{E},\;X_{g}L_{g}$ and $X_{gL}\left(L_{E}⊗L_{g}\right)$, which belong to environments, genotypes and GE components, respectively. We first computed the design matrices (dummy variables) of environments $\left(X_{L}\right)$, genotypes $\left(X_{g}\right)$ and GE interaction $\left(X_{gL}\right)$. Then, $L_{g}$ and $L_{E}$ were computed. $L_{g}$ denotes the square root of the genomic relationship matrix (G), whereas $L_{E}$ denotes the square root of environmental relationship matrix H. To compute matrix H, we used the environmental covariates when available, as in datasets 1 (Indica) and 2 (Japonica), whereas for the remaining datasets for which environmental covariables were not collected, we computed only the design matrix of environments $\left(X_{L}\right)$. Furthermore, both MT models (GBLUP and PLS) were implemented with R statistical software, whereas the MT- GBLUP model was implemented with BGLR library [25] and the MT-PLS model was implemented with the pls library [26].

### 2.2. Datasets

#### 2.2.1. Dataset 1: Indica

This dataset contains information on the phenotypic performance of 4 traits (GY = grain yield, PHR = percentage of head rice recovery, GC = percentage of chalky grain, PH = plant height) of rice and was reported by Monteverde et al. [27] for 3 environments (years 2010, 2011 and 2012). The number of genotypes evaluated each year (environment) was 327, and each year, environmental covariates were measured in three stages (one for each developmental stage: maturation, reproduction and vegetation) for each of the following 18 environmental covariates: (1) ThermAmp denotes average of daily thermal amplitude calculated as max temperature (°C)—min temperature (°C); (2) RelSun denotes the relative sunshine duration (%) computed as the quotient between the real duration of the brightness of the sun and the possible geographical or topographic duration; (3) SolRad denotes solar radiation (cal/cm^2^/day) calculated using Armstrong’s formula; (4) EfPpit denotes effective precipitation (mm) computed as the average of daily precipitation in mm that is actually added and stored in the soil; (5) DegDay denotes the mean of daily average temperature minus 10°; (6) RelH denotes relative humidity (hs) computed as the sum of daily hours (0–24 h) with a relative humidity equal to 100%; (7) PpitDay denotes the precipitation day computed as the sum of days during which it rained; (8) MeanTemp denotes the mean of temperature (°C) over 24 h (0–24 h); (9) AvTemp denotes the average temperature (°C) calculated as daily (Max + Min) / 2; (10) MaxTemp denotes the average maximum daily temperature (°C); (11) MinTemp denotes the average minimum daily temperature (°C); (12) TankEv denotes tank water evaporation (mm) computed as the amount of evaporated water under the influence of sun and wind; (13) Wind denotes wind speed (2 m/km/24 h) computed as the distance covered by wind (in km) over 2 m height in one day; (14) PicheEv denotes Piche evaporation (mm) computed as the amount of evaporated water without the influence of the sun; (15) MinRelH stands for the minimum relative humidity (%) computed as the lowest value of relative humidity for the day; (16) AccumPpit denotes the daily accumulated precipitation (mm); (17) Sunhs denotes sunshine duration computed as the sum of total hours of sunshine per day; and (18) MinT15 denotes the minimum temperature below 15° computed as the sum of the days when the minimum temperature was below 15. More details related to how these environmental covariates were measured can are presented by Monteverde et al. [27].

The total number of assessments in this balanced dataset is 981, as each line is included once in each environment. The genotyping-by-sequencing (GBS) markers datasets were filtered to retain markers, with 50% missing data after imputation and a minor allele frequency (MAF) > 0.05. The markers remaining after quality control were 92,430 SNPs for each line and are coded as 0, 1 and 2.

#### 2.2.2. Dataset 2: Japonica

Monteverde et al. [27] investigated this rice dataset that belongs to the tropical Japonica population with the same four traits (GY = grain yield, PHR = percentage of head rice Recovery, GC = percentage of chalky grain, PH = plant height) as the Indica population (dataset 1) but over the course of 5 years (2009, 2010, 2011, 2012 and 2013). A total of 93, 292, 316, 316 and 134 genotype lines were evaluated for years 2009, 2010, 2011, 2012 and 2013, respectively. The same 54 environmental covariates studied in the Indica dataset (Dataset 1) were investigated in this dataset. In this dataset, a total of 1051 assessments were evaluated in the five years. In this dataset, 320 genotypes were evaluated, with 44,598 markers remaining for each line after quality control, coded with 0, 1 and 2.

#### 2.2.3. Dataset 3: Groundnut Data

This dataset was investigated by Pandey et al. [28] and contains phenotypic and genotypic information for 318 genotypes and four environments. In the present study, we assessed the prediction accuracy of the following four traits: seed yield per plant (SYPP), pods per plant (NPP), pod yield per plant (PYPP) and yield per hectare (YPH). The environments were designated as Aliyarnagar_Rainy 2015 (ENV1), Jalgoan_Rainy 2015 (ENV2), ICRISAT_Rainy 2015 (ENV3) and ICRISAT Post-Rainy 2015 (ENV4).

This dataset contains a total of 1272 assessments and is balanced, as each genotype is included once in each environment. For each genotype, 8,268 single-nucleotide polymorphism (SNP) markers (coded with 0, 1 and 2) were available after quality control.

#### 2.2.4. Dataset 4: Disease Data

In this dataset with 438 wheat genotypes (lines), three traits (diseases) were investigated: *Pyrenophora tritici-repentis* (PTR); *Parastagonospora nodorum* (SN), a major fungal pathogen of the wheat fungal taxon; and *Bipolaris sorokiniana* (SB), which causes seedling diseases, common root rot and spot blotch of several crops, such as barley and wheat. These 438 lines were evaluated over a long period of time in a greenhouse for six replicates. The replicates were considered environments (Env1, Env2, Env3, Env4, Env5 and Env6). For the three evaluated traits, the total number of observations was 438 × 6 = 2628.

DNA samples were genotyped using 67, 436 SNPs. For each marker, the genotype for each line was coded as the number of copies of a designated marker-specific allele carried by the line (absence = zero and presence = one). SNP markers with unexpected heterozygous genotypes were recoded as either AA or BB. Markers with more than 15% missing values were removed, as well as markers with MAF < 0.05. A total of 11,617 SNPs were still available for analysis after quality control and imputation.

#### 2.2.5. Datasets 5–6: Elite Wheat Yield Trial (EYT) Years 2013–2014 and 2014–2015

These two datasets were collected by the Global Wheat Program (GWP) of the International Maize and Wheat Improvement Center (CIMMYT) and belong to elite yield trials (EYT) established in four cropping seasons with four or five environments. Dataset 5 is from 2013–2014, and Dataset 6 is from 2014–2015. EYT datasets 5 and 6 contain 776 and 775 genotypes, respectively. An alpha-lattice experimental design was implemented, and the lines were sown in 39 trials, each covering 28 lines and 2 checks in 6 blocks with 3 replications. Several traits were available for some environments and genotypes in each dataset. In this study, we included four traits measured for each line in each environment: days to heading (DTHD, number of days from germination to 50% spike emergence), days to maturity (DTMT, number of days from germination to 50% physiological maturity or the loss of the green color in 50% of the spikes), plant height (PH) and grain yield (GY). For full details of the experimental design and how the best linear unbiased estimates (BLUEs) were computed, see Juliana et al. [29].

The lines examined in dataset 6 were investigated in five environments, whereas dataset 5 was investigated in four environments. For EYT **dataset 5**, the environments were bed planting with five irrigations (Bed5IR), early heat (EHT), flat planting with five irrigations (Flat5IR) and late heat (LHT). For EYT **dataset 6**, the environments were bed planting with two irrigations (Bed2IR), Bed5IR, EHT, Flat5IR and LHT.

Genome-wide markers for the 1551 (776 + 775) genotypes in the two datasets were obtained using genotyping by sequencing (GBS) [30,31] at Kansas State University on an Illumina HiSeq2500. After filtering, 2038 markers were obtained from an initial set of 34,900 markers. Missing marker data were imputed using LinkImpute [32] and implemented in TASSEL (Trait Analysis by Association Evolution and Linkage) version 5 [33]. Genotypes with more than 50% missing data were removed, and 2515 genotypes were used in this study (776 lines in the first dataset and 775 lines in the second dataset).

### 2.3. Metrics for Evaluation of Prediction Accuracy

We implemented sevenfold cross validation for each of the 6 datasets [8]. Therefore, we randomly divided the dataset into 7 subsets of similar size, using 7−1=6 subsets as a training set and the remaining group as a test set until each of the 7 subsets played the role of test set once. In the case of the Bayesian model (Model 1), we did not require a tuning process, but in the case of PLS and random forest, we divided the respective training set into an inner training set (80% of the training set) and a validation set (20% of the training set). This nested cross validation was implemented under 5-fold cross validation. Then, the average of the five validation sets was reported as the accuracy of the predictions to select the optimal hyperparameters, which, in the case of PLS, was the number of principal components that must be retained. In the case of random forest, the number of trees and the node size were the tuned hyperparameters. In this case of random forest, the optimal hyperparameters were selected with the Bayesian optimization approach. Bayesian optimization is a sequential design approach for the optimization of black-box functions that do not assume any functional forms. It is usually used to optimize functions that are difficult to evaluate. Because a known objective function is not assumed, the Bayesian approach considers a random function; therefore, a prior is considered to incorporate beliefs about the behavior of the function. After collecting the function, these are treated as data, and the prior is adjusted to form the posterior distribution over the objective function. Then, the posterior distribution is used to establish an acquisition function that determines the next query points, and this step is repeated until an optimal set of hyperparameters is found [34]. Subsequently, with these optimal hyperparameters, we refitted the model using the complete training set (information of 6 folds); finally, with these refitted models, the predictions of the test were obtained.

As a metric to evaluate the prediction error, we used the average of the seven folds of the normalized mean square error ($NRMSE=\frac{1}{n}\overset{n}{\underset{j=1}{\sum }}N\;RMSE_{V_{j}}=\frac{1}{n}\overset{n}{\underset{j=1}{\sum }}\frac{RMSE_{V_{j}}}{{\overset{-}{y}}_{V_{j}}}$), where n is the number of observations in the testing set; $RMSE_{V_{j}}=\sqrt{\frac{1}{T}\overset{T}{\underset{i=1}{\sum }}{\left(y_{i}-\hat{f}\left(x_{i}\right)\right)}^{2}}$ denotes the root of the mean square error of variable j; $y_{i}$ denotes the observed value of i; and $\hat{f}\left(x_{i}\right)$ denotes the predicted value for observation i, where i=1,⋯,n. This metric was calculated for the GBLUP ($NRMSE_{GBLUP}$), random forest ($NRMSE_{RF}$) and PLS ($NRMSE_{PLS}$) models; then, taking the GBLUP model as a reference, we calculated the relative efficiency of the other two models. Therefore, we computed the relative efficiencies as:$$RE_{RF}=\frac{NRMSE_{GBLUP}}{NRMSE_{RF}}\;RE_{PLS}=\frac{NRMSE_{GBLUP}}{NRMSE_{PLS}}$$

When $RE_{RF}>1$ ($RE_{PLS}>1$), the best performance of predictions was obtained by using random forest model (PLS); however, when $RE_{RF}<1$ ($RE_{PLS}<1$), the GBLUP model was superior in terms of prediction error. When $RE_{RF}=1$ ($RE_{PLS}=1$), the two methods were equally efficient.

### 2.4. Functions for Implementing the Multi-Trait Models Using the SKM Library

The main functions of the SKM library [18] with respect to implementation of the three machine learning models were:

***bayesian_model()***: is a wrapper of the *BGLR::BGLR()* and *BGLR::Multitrait()* functions, the latter being the function used to fit a multivariate Bayesian regression model. The main arguments used to adjust this model are *x*, *y* and *testing_indices*, with which we specify the information of the predictor variables, response variables and indices for the testing set, respectively. Unlike the other functions used to implement the seven machine learning algorithms offered by the SKM library, it is necessary to specify the indices of the training set. The *x* argument must be a list of nested lists, wherein each list represents an effect of the predictor. To implement the GBLUP model in its Bayesian form, it is necessary to specify this argument as:
*x* = *list*(*G* = *list*(*x* = ***G***, *model* = “*BGBLUP*”)), 
where ***G*** denotes the genomic relationship matrix.Use *help(“bayesian_model”)* in the *R* console to see more details about the parameters of this function.***partial_least_squares()***: is a wrapper of the pls::*plsr()* function, which is the function used to fit a multivariate partial least squares regression model for numerical responses. The main arguments used to fit this model are *x* and *y*, with which we specify the predictor variables and response variables, respectively. This function is also useful for implementing single-trait prediction models. Use *help(“partial_least_squares”)* in the *R* console to see more details about the parameters of this function.***random_forest()***: is a wrapper of the *randomForestSRC::rfsrc()* function, which is the function used to fit a random forest model. The main arguments used to fit this model are:
➢*x*: predictor (or independent) variables in matrix form;➢*y*: response variables (or dependent) variables in a matrix or in a *data frame (in the multivariate case)* or *vector* (in the univariate case);➢***trees_number***: is a **tunable** hyperparameter that specifies the number of regression trees used;➢***node_size***: is a **tunable** hyperparameter that specifies the minimum number of terminal nodes in each regression tree;➢***tune_type***: is an argument that specifies the type of tuning to use for hyperparameters (“*Grid_search*” by default). In the case of the “*Grid_search*” tuning type, the proposed values for the hyperparameters must be specified through a vector, whereas in the case of the “*Bayesian_optimization*” method, they must be specified through a list of two elements that indicate the range of the proposed values for each hyperparameter.

To implement the RF model with the datasets, we used Bayesian optimization to tune the hyperparameters (*tune_type = “Bayesian_optimization”*), values between 5 and 50 for the number of regression trees (*trees_number = list(min = 5, max = 50)*) and values between 5 and 15 for the number of terminal nodes (*node_size = list(min = 5, max = 15)*). 

Use *help(“partial_least_squares”)* in the *R* console to view more details about the parameters of this function.

On the other hand, to train the model and evaluate its predictive capacity, it is necessary to divide the dataset into two sets (the training set and the testing set), depending on the cross-validation scheme. For this task, the SKM library offers two general-purpose functions:***cv_random()***: generates the folds to use under a random cross-validation framework, and we need to provide as input the number of observations (*records_number*), the number of folds (*folds_number*) and the proportion of observations to be included in the test set (*testing_proportion*). Each fold is built using random sampling with replacement and the proportion specified for the test set and the rest for the training set.***cv_kfold()***: generates the folds to be used under the k-fold cross-validation approach; the number of observations (*records_number*) and the number of desired folds (*k*) must be provided as the input. If the number specified in the *records_number* argument corresponds to the number of environments and the k argument corresponds to the number of environments, then the cross-validation scheme corresponds to leave-one-environment-out (LOEO).

Finally, ***gs_summaries()*** is a function that helps to evaluate the predictive capacity of each model by reporting summary statistics of the predictions made in the various generated folds. The main argument required in this function is a data frame containing the following columns: *Fold, Line, Env, Observed* and *Predicted*; the output of this function is a list of the prediction performance with three summaries (by “line”, by “env” and by “fold”). Use *help(“gs_summaries”)* in the *R* console to see more details about this feature. The SKM library can be installed from GitHub with the following lines of code:devtools::install_github (“cran/ randomForestSRC”)devtools::install_github (“gdlc/BGLR-R”)devtools::install_github (“rstudio/ tensorflow”)if (!require (“devtools”)) {install.packages (“devtools”)}devtools::install_github (“brandonmosqueda/SKM”)

### 2.5. Data Availability and Appendix A

The genomic and phenotypic data for the six datasets (datasets 1–6) included in this study are available at https://hdl.handle.net/11529/10548728. Also, this link includes Appendix A with Figures and Tables for datasets 3–6, as well as R codes for SKM.

## 3. Results

The results are presented in two subsections corresponding to the first two datasets, dataset 1 (Indica), and dataset 2 (Japonica), and 5 sub-subsections corresponding to the variance components and heritability’s plus the results in terms of the prediction accuracy of the four predictors under study: (1) G predictor, (2) G + E predictor, (3) E + G + GE predictor and (4) G + E predictor. As previously mentioned, results from datasets 3–6 are presented in Appendix A with figures, tables and R codes for SKM are also available at https://hdl.handle.net/11529/10548728. We did not compare the time required for implementation between the three methods because, in general, it is known that the most demanding method in terms of computational resources is the random forest model, and the least demanding is the GBLUP method.

### 3.1. Dataset 1: Indica

Data are shown in Figure 1 and in Table 1 and Table 2.


*
**Heritability and variance components**
*


Table 1 shows that the lowest heritability for dataset 1 (Indica) is observed for the GY trait (0.47), with the highest heritability for trait PH (0.76). With respect to the GY trait, the highest and lowest variance components correspond to locations (496020.35) and residual (336143.27), respectively. For trait PH, the highest and lowest variance components correspond to the effects of hybrids (9.66) and the hybrids × location interaction (1.96), respectively (Table 1).


*
**With predictor = G**
*


When only the genotypic information was considered in the predictor, under the **sevenfold CV cross-validation** scheme, we observed that the relative efficiencies of the **GBLUP** model vs. the **PLS** model were 0.909, 0.964 and 0.916 for environments (years) 2010, 2011 and 2012, respectively; that is, in each of the environments, the performance of the predictions of the **PLS** model was lower compared to that of the **GBLUP** model, and the loss in the accuracy of the predictions was 9.1% (2010), 3.6% (2011) and 8.4% (2012). In addition, across all environments (global), we observed that the **GBLUP** model achieved better performance than the **PLS** model, as the relative efficiency was equal to 0.928; that is, across all environments, the **GBLUP** model outperformed the **PLS** model by 7.2%, as $\frac{1}{RE\_PLS}=\frac{1}{0.928}=1.078$ (Figure 1 with predictor = G) (Table 2).

Similarly, we observed that the relative efficiencies of the **GBLUP** model vs. the **random forest** model, under the **sevenfold CV cross-validation** scheme, were 0.984, 0.994 and 0.949 for years 2010, 2011 and 2012, respectively; that is the performance of the **random forest** model was 1.6% (2010), 0.6% (2011) and 5.1% (2012) lower than that of the **GBLUP** model. In addition, across all environments (global), we observed that the **GBLUP** model performed better than the random forest model, as the relative efficiency was equal to 0.978; that is, across all environments, the **GBLUP** model outperformed the **random forest** model by 2.2% (Figure 1 with predictor = G) (Table 2).


*
**With predictor = G + E**
*


When only the effect of environments and genotypic information on the predictor was considered, under the **sevenfold CV cross-validation** scheme, we observed that the relative efficiencies of the **GBLUP** model vs. the **PLS** model were 0.904, 0.962 and 0.931 for environments (years) 2010, 2011 and 2012, respectively; that is, in each of the environments, the performance of the predictions of the **PLS** model was lower than that of the **GBLUP** model, as the loss in the accuracy of the predictions was 9.6% (2010), 3.7% (2011) and 6.9% (2012). In addition, across all environments (global), we observed that the **GBLUP** model performed better than the **PLS** model, as the relative efficiency was equal to 0.925; that is, across all environments, the **GBLUP** model outperformed the **PLS** model by 8.1%, as $\frac{1}{RE\_PLS}=\frac{1}{0.925}=1.081$ (Figure 1 with predictor = G + E) (Table 2).

With respect to the relative efficiencies of the **GBLUP** model vs. the **random forest** model, under the **sevenfold CV cross-validation** scheme, we observed that the relative efficiencies of the **GBLUP** model vs. the **random forest** model were 1.027, 1.027 and 1.009 for environments (years) 2010, 2011 and 2012, respectively; that is, in each of the environments (years), the performance of the predictions of the **random forest** model was higher than that of the **GBLUP** model, and the gains in the accuracy of the predictions were 2.7% (2010), 2.7% (2011) and 0.9% (2012). In addition, across all environments (global), we observed that the **random forest** model performed better than the **GBLUP** model, as the relative efficiency was equal to 1.009; that is, across all environments, the **GBLUP** model was surpassed by the **random forest** model by 0.9% (Figure 1 with predictor = G + E) (Table 2).


*
**With predictor = E + G + GE**
*


When GE, which contains genotypic information about the interaction with the environments, was considered in the predictor, under the **sevenfold CV cross-validation** scheme, we observed that the relative efficiencies of the **GBLUP** model vs. the **PLS** model were 0.898, 0.934 and 0.944 for the environments (years) 2010, 2011 and 2012, respectively; that is, in each of the environments, the performance of the predictions of the **PLS** model was lower compared to that of the **GBLUP** model and the loss in the accuracy of the predictions was 10.2% (2010), 6.6% (2011) and 5.6% (2012). In addition, across all environments (global), we observed that the **GBLUP** model performed better than the **PLS** model, as the relative efficiency was equal to 0.918; that is, across all environments, the **GBLUP** model outperformed the **PLS** model by 8.9%, as $\frac{1}{RE\_PLS}=\frac{1}{0.918}=1.089$ (Figure 1 with predictor = G + E + GE) (Table 2).

Similarly, with respect to the relative efficiencies of the **GBLUP** model vs. the **random forest** model, under the **sevenfold CV cross-validation** scheme, we observed that the relative efficiencies of the **GBLUP** model vs. the **random forest** model were 0.991, 0.982 and 0.990 for environments (years) 2010, 2011 and 2012, respectively; that is, in each of the environments (years), the performance of the predictions of the **random forest** model was lower than that of the **GBLUP** model, as the losses in the accuracy of the predictions were 0.9% (2010), 1.8% (2011) and 1% (2012). In addition, across all environments (global), we observed that the **GBLUP** model performed better than the **random forest** model, as the relative efficiency was equal to 0.976; that is, across all environments, the **random forest** model was outperformed by the **GBLUP** model by 2.4% (Figure 1 with predictor = G + E + GE) (Table 2).


*
**With predictor = G + GE**
*


When the effect of environments on the predictor was not considered and under the **sevenfold CV cross-validation** scheme, we observed that the relative efficiencies of the **GBLUP** model vs. the **PLS** model were 0.913, 0.895 and 0.931 for environments (years) 2010, 2011 and 2012, respectively; that is, in each of the environments, the performance of the predictions of the **PLS** model was lower compared to that of the **GBLUP** model, and the loss in the accuracy of the predictions was 8.7% (2010), 10.5% (2011) and 6.9% (2012). In addition, across all environments (global), we observed that the **GBLUP** model performed better than the **PLS** model, as the relative efficiency was equal to 0.907; that is, across all environments, the **PLS** model was surpassed by the **GBLUP** model by 9.3% (Figure 1 with predictor = G + GE) (Table 2).

Regarding the relative efficiencies of the **GBLUP** model vs. the **random forest** model, under the **sevenfold CV cross-validation** scheme, we observed that the relative efficiencies of the **GBLUP** model vs. the **random forest** model were 0.996, 0.983 and 0.990 for environments (years) 2010, 2011 and 2012, respectively; that is, in each of the environments (years), the performance of the predictions of the **random forest** model was lower than that of the **GBLUP** model, and the loss in the accuracy of the predictions was 0.4% (2010), 1.7% (2011) and 1% (2012). In addition, across all environments (global), we observed that the **GBLUP** model performed better than the **random forest** model, as the relative efficiency was equal to 0.980; that is, across all environments, the **random forest** model was outperformed by the **GBLUP** model by 2% (Figure 1 with predictor = G + GE) (Table 2).

### 3.2. Dataset 2: Japonica

Data are shown in Figure 2 and in Table 3 and Table 4.


*
**Heritability and variance components**
*


Table 3 shows the variance components and heritabilities of each trait of the Japonica dataset (2). The lowest heritability for this dataset was observed with respect to the GC trait (0.25), whereas the highest heritability was achieved with respect to the PH trait (0.62). For the GC trait, the highest and lowest variance components correspond to the effects of locations (0.006) and the hybrids × location interaction (0.000), respectively. On the other hand, for trait PH, the highest and lowest variance components correspond to the effects of locations (35.950) and the hybrids × location interaction (0.002), respectively (Table 3).


*
**With predictor = G**
*


When only the genotypic information was considered in the predictor, under the **sevenfold CV cross-validation** scheme, we observed that the relative efficiencies of the **GBLUP** model vs. the **PLS** model were 0.983, 1.003, 1.000, 1.057 and 0.937 for environments (years) 2009, 2010, 2011, 2012 and 2013, respectively; that is, only in environments (years) 2010 and 2012, the performance of the **PLS** regression predictions was higher than that of the **GBLUP** model, as the accuracy of the predictions was 0.3% (2010) and 5.7% (2012), whereas the prediction performance was the same in environment (year) 2011 for both models. However, across all environments (global), we observed that the **GBLUP** model performed better than the **PLS** model, as the relative efficiency was equal to 0.973; that is, across all environments, the **GBLUP** model outperformed the **PLS** model by 2.7% (Figure 2 with predictor = G) (Table 4).

Regarding to the relative efficiencies of the **GBLUP** model vs. the **random forest** model, under the **sevenfold CV cross-validation** scheme, we observed that the relative efficiencies of the **GBLUP** model vs. the **Random Forest** model were 0.884, 0.951, 0.934, 0.967 and 0.932 for environments (years) 2009, 2010, 2011, 2012 and 2013, respectively; that is, in each of the environments, the performance of the predictions of the **random forest** model was lower compared to that of the **GBLUP** model, and the losses in the accuracy of predictions were 11.6% (2009), 4.9% (2010), 6.6% (2011), 3.3% (2012) and 6.8% (2013). In addition, across all environments (global), we observed that the **GBLUP** model performed better than the **random forest** model, as the relative efficiency was equal to 0.906; that is, across all environments, the **GBLUP** model outperformed the **random forest** model by 10.4%, as $\frac{1}{RE\_RF}=\frac{1}{0.906}=1.104$ (Figure 1 with predictor = G).


*
**With predictor = G + E**
*


When only the effect of environments and genotypic information on the predictor was considered, under the **sevenfold CV cross-validation** scheme, we observed that the relative efficiencies of the **GBLUP** model vs. the **PLS** model were 0.938, 0.954, 0.864, 0.904 and 0.803 for environments (years) 2009, 2010, 2011, 2012 and 2013, respectively; that is, in each of the environments, the performance of the predictions of the **PLS** model was lower compared to that of the **GBLUP** model, as the loss in the accuracy of the predictions was 6.2% (2009), 4.6% (2010), 3.6% (2011), 9.6% (2012) and 19.7% (2013). In addition, across all environments (global), we observed that the **GBLUP** model performed better than the **PLS** model, as the relative efficiency was equal to 0.823; that is, across all environments, the **GBLUP** model outperformed the **PLS** model by 21.5% $\frac{1}{RE\_PLS}=\frac{1}{0.823}=1.215$ (Figure 2 with predictor = G + E) (Table 4).

Similarly, with respect to the relative efficiencies of the **GBLUP** model vs. the **random forest** model, under the **sevenfold CV cross-validation** scheme, we observed that the relative efficiencies of the **GBLUP** model vs. the **random forest** model were 0.977, 1.112, 0.928, 0.916 and 0.859 for environments (years) 2009, 2010, 2011, 2012 and 2013, respectively; that is, only in environment (year) 2010 was the performance of the predictions of the **random forest** model superior with to that of the **GBLUP** model, and the gain in the accuracy of predictions was 11.2% (2013). However, across all environments (global), we observed that the **GBLUP** model performed better than the **random forest** model, as the relative efficiency was equal to 0.857; that is, across all environments, the **GBLUP** model outperformed the **random forest** model by 14.3% (Figure 2 with predictor = G + E) (Table 4).


*
**With predictor = E + G + GE**
*


When GE, which contains genotypic information on the interaction with environments, was also considered in the predictor, under the **sevenfold CV cross-validation** scheme, we observed that the relative efficiencies of the **GBLUP** model vs. the **PLS** model were 0.943, 0.812, 0.862, 0.919 and 0.800 for environments (years) 2009, 2010, 2011, 2012 and 2013, respectively; that is, in each of the environments, the performance of the predictions of the **PLS** model was lower compared to that of the **GBLUP** model, and the loss in the accuracy of the predictions was 5.7% (2009), 18.8% (2010), 13.8% (2011), 9.6% (2012) and 20.0% (2013). In addition, across all environments (global), we observed that the **GBLUP** model performed better than the **PLS** model, as the relative efficiency was equal to 0.828; that is, across all environments, the **GBLUP** model outperformed the **PLS** model by 17.2% (Figure 2 with predictor = G + E + GE) (Table 4).

With respect to the relative efficiencies of the **GBLUP** model vs. the **random forest** model, under the **sevenfold CV cross-validation** scheme, we observed that the relative efficiencies of the **GBLUP** model vs. the **random forest** model were 1.024, 0.939, 0.928, 0.922 and 0.851 for environments (years) 2009, 2010, 2011, 2012 and 2013, respectively; that is, only in environment (year) 2009 was the performance of the predictions of the **random forest** model superior to that of the **GBLUP** model, as the gain in the accuracy of predictions was 2.4% (2013). In addition, across all environments (global), we observed that the **GBLUP** model performed better than the **random forest** model, as the relative efficiency was equal to 0.859; that is, across all environments, the **random forest** model was outperformed by the **GBLUP** model by 14.1% (Figure 2 with predictor = G + E + GE) (Table 4).


*
**With predictor = G + GE**
*


When E, the effect of the environments on the predictor, was not considered and under the **sevenfold CV cross-validation** scheme, we observed that the relative efficiencies of the **GBLUP** model vs. the **PLS** model were 0.673, 0.473, 0.767, 0.675 and 0.684 for environments (years) 2009, 2010, 2011, 2012 and 2013, respectively; that is, in each of the environments, the performance of the predictions of the **PLS** model was lower compared to that of the **GBLUP** model, and the loss in the accuracy of the predictions was 32.7% (2009), 52.7% (2010), 23.3% (2011), 32.5% (2012) and 31.6% (2013). In addition, across all environments (global), we observed that the **GBLUP** model performed better than the **PLS** model, as the relative efficiency was equal to 0.663; that is, across all environments, the **PLS** model was outperformed by the **GBLUP** model by 33.7% (Figure 2 with predictor = G + GE).

With respect to the relative efficiencies of the **GBLUP** model vs. the **random forest** model, under the **sevenfold CV cross-validation** scheme, we observed that the relative efficiencies of the **GBLUP** model vs. the **random forest** model were 1.072, 0.968, 0.921, 0.922 and 0.833 for environments (years) 2009, 2010, 2011, 2012 and 2013, respectively; that is, only in environment (year) 2009 was the performance of the predictions of the **random forest** model superior to that of the **GBLUP** model, and the gain in the accuracy of predictions was 7.9% (2009). In addition, across all environments (global), we observed that the **GBLUP** model performed better than the **random forest** model, as the relative efficiency was equal to 0.859; that is, across all environments, the **random forest** model was outperformed by the **GBLUP** model by 14.1% (Figure 2 with predictor = G + GE) (Table 4).

## 4. Discussion

In a recent study, Montesinos-López et al. [35] investigated the partial least square (PLS) regression methodology for the prediction of one full environment of a single trait (ST) and compared its prediction performance with that of the GBLUP method. In all datasets, the ST-PLS method outperformed the ST-GBLUP method by margins between 0% and 228.28% across traits, environments, and types of predictors. Furthermore, the multi-trait partial least square (MT-PLS) regression method can model complex biological events, is flexible in consideration of various factors and is unaffected by data collinearity. Montesinos-López et al. [36] showed that the MT-PLS model is valuable for improving genomic prediction of high-dimensional plant breeding data, as it can model multiple responses and efficiently deal with multicollinearity. MT-PLS explicitly uses the correlation structure among such traits.

MT genomic prediction models allow breeders to save significant phenotyping resources and increase the prediction performance of unobserved target traits by exploiting accessible information from non-target or auxiliary (secondary) traits. Therefore, these models are attractive and promising for MT prediction in genomic selection. However, due to the lack of accessible software for multi-trait models, it is not easy for users to implement such models without a strong background in computation and programming. Therefore, in this research, with the goal of facilitating the widespread use of multi-trait models by scientists for genomic selection, we illustrated with R code how to implement MT-GBLUP, MT-random forest and MT-PLS in the SKM library [18].

Additional details with respect to the implementation of the three models are provided in Appendix A, and with a minimum modification, these codes can be used by users to implement these models with their own data. Additionally, the SKM library allows enables design of cross-validation strategies (cv_random and cv_kfold) with simple commands that are in agreement with real prediction scenarios of interest for breeders. Additionally, the SKM library offers some functions for summary of the prediction accuracy (numeric_summary(), categorical_summary() and gs_summaries()), as well as many options for metrics (for example, Pearson’s correlation (Cor), mean square error (MSE), mean absolute error (MAE), root mean square error (RMSE), normalized root mean square error (NRMSE), coefficient of determination (R2) and mean arctangent absolute percentage error (MAAPE)) for continuous response variables and for categorical response variables (proportion of cases correctly classified (PCCC), kappa coefficient (Kappa), Brier score, sensitivity and specificity) [8].

We found that in most datasets, the best predictions under the three models were achieved under two of the four predictors (E + G and E + G + GE), with the worst predictions achieved under predictors G and G + GE, which means that including the environment (E) effect is of paramount importance to improve the prediction performance. Therefore, as pointed out by one of the reviewers, using non-linear kernels with the climate covariates could improve the genomic prediction, as these kernels are more powerful in terms of capturing non-linear patterns in the data [37,38]. However, this was not evaluated in the present research. Additionally, in most of the datasets analyzed in this paper, we did not observe a strong improvement in terms of prediction accuracy as a result of including the genotype × environment interaction, which is, in part, due to a lack of a strong genotype × environment interaction. In general, the same patterns observed in the Indica and Japonica datasets were observed in the other four datasets (Groundnut, Disease, EYT_1 and EYT_2), with more specific details of the prediction performance under the four predictors are available in the Appendix A. We also observed that in general, the MT-GBLUP model turned out to be superior to the other two MT models (PLS and random forest) under the first two predictors (G, E + G). However, under predictors three (E + G + G × E) and four (G + G × E), the random forest model achieved the best performance. The satisfactory performance of the random forest model can be attributed to two important facts: one is that it naturally captures non-linear patterns in the data, and the other is that for the training process, we used Bayesian optimization.

Bayesian optimization using the Bayes theorem to direct a search for an effective and efficient global optimization problem. It works by building a stochastic model of the objective function (surrogate function), which is then explored efficiently with an acquisition function; candidate samples were previously selected for evaluation of the real objective function. Bayesian optimization maximizes the function with a few evaluations by evaluating the next point according to the previous observation. In probability terms, in order to determine the next evaluation, given the current evaluations as a starting point for the next evaluation, Bayesian optimization uses posterior distributions. Acquisition functions include (1) heuristics for evaluating the utility of a point, (2) functions of the surrogate posterior, (3) tools to combine exploration and exploitation and (4) tools that are inexpensive to evaluate. For these reasons, Bayesian optimization is usually preferred over grid search when the functions are expensive to evaluate and will take a long time for evaluation, which makes grid search impractical.

Although the random forest model can capture non-linear patterns better than the MT-GBLUP and MT-PLS model, it does not efficiently capture the degree of correlation between traits, as the loss function in the random forest model does not take into account (does not estimate) a covariance matrix between traits, as is the case for the MT-GBLUP model [10,11]. Therefore, it is expected that when the degree of correlation between traits is low and the degree of non-linear patterns is high, the MT random forest model will outperform the MT-GBLUP method, whereas when the opposite situation occurs, the MT-GBLUP model will outperform the MT random forest method. Because both factors (degree of correlation between traits and degree of non-linear patterns) are important, robust statistical machine learning methods must be used to take advantage of these factors to increase the prediction accuracy.

We also observed that the best (worst) predictions were associated (for each dataset) with traits with higher (lower) heritability, which is expected, as there is evidence of a clear relationship between prediction accuracy and heritability. However, because we only reported the prediction performance across traits, other details could not be fully observed in the provided tables and figures. Furthermore, under the implemented sevenfold CV, we observed that in general, the MT-PLS model achieved slightly worse performance than MT-GBLUP and MT random forest models; however, this finding cannot be extrapolated to all types of cross-validation strategies because evidence suggests that when the goal is to predict a complete environment (or year), the ST-PLS and MT-PLS models outperform the ST-GBLUP and MT-GBLUP models by considerable margins [34,35], which can be attributed, in part, to the fact that the ST and MT-PLS models first involve a variable selection process during which a considerable amount noise is discarded, and at the end, the training process applied with a latent variable of inputs.

The goal of this study was to formally compare genomic prediction accuracy between MT prediction models and ST prediction models. The authors of many publications have performed such a comparison, showing that when the degree of correlation is moderate or high between traits, MT models outperform ST models in terms of genomic prediction accuracy [9,10,11].

## 5. Conclusions

In this study, the prediction performance of three multi-trait statistical machine learning methods was compared; we found that the three investigated models are competitive. However, f under predictors E + G and E + G + G × E, the random forest performed the best of the three investigated models. Under predictors one (G) and two (E + G), the MT-GBLUP model achieved slightly better performance than the other two models. Our empirical results show that any of the three models is competitive in terms of predictions in the context of multi-trait and multi-environment data and that in some cases, one of these models outperforms the other two. We also illustrated the availability of these three multi-trait methods in the SKM library, which, in addition to many single-trait statistical machine learning methods, allows for the implementation of multi-trait methods. With the provided code, implementing multi-trait methods under the SKM library is very easy; therefore, any user can successfully use this library correctly. It is important to point out that our empirical findings are only valid for the type of cross validation called k-fold cross validation, as only this type of cross validation was implemented in the present study; we intend to perform more research to support our findings and determine whether the investigated models are valid for other strategies of cross validation.

## Figures and Tables

**Figure 1 genes-13-01494-f001:**
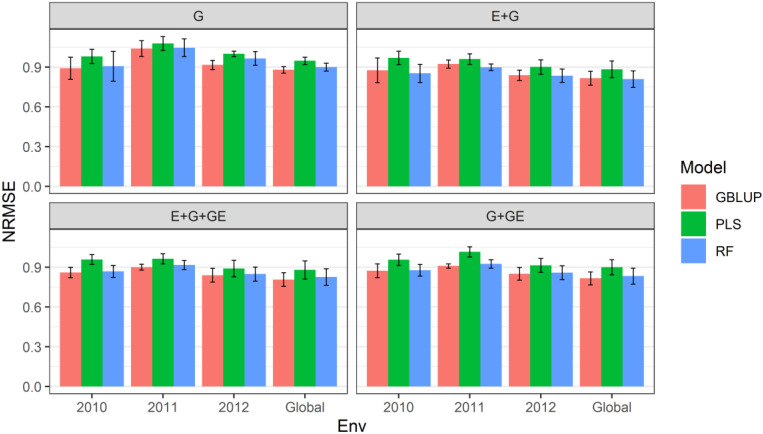
Prediction performance for each environment and across environments (Global) of dataset 1 (Indica) in terms of normalized mean square error (NRMSE) under four predictors (G, genotypic information; E + G. environment plus genotypic information; E + G + GE, environment plus genotypic plus genotype by environment interaction information; and G + GE, genotypic plus genotype by environment interaction) and under the sevenfold cross-validation (CV) scheme.

**Figure 2 genes-13-01494-f002:**
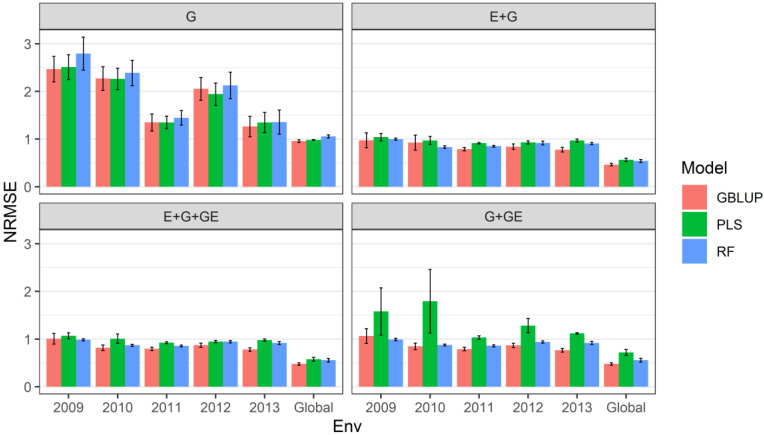
Prediction performance for each environment and across environments (global) of dataset 2 (Japonica) in terms of normalized mean square error (NRMSE) under four predictors (G, genotypic information; E + G, environment plus genotypic information; E + G + GE, environment plus genotypic plus genotype by environment interaction information; and G + GE, genotypic plus genotype by environment interaction) and under the sevenfold cross-validation (CV) scheme.

**Table 1 genes-13-01494-t001:** Variance components (variance) and heritability estimates for **dataset 1** (**Indica**) for each trait. CV denotes coefficient of variation, and Locs denotes the average number of locations.

Trait	Component	Variance	Heritability	CV	Locs
GY	Loc:Hybrid	394520.80	0.47	0.11	3
GY	Hybrid	361259.05	0.47	0.11	3
GY	Loc	496020.35	0.47	0.11	3
GY	Residual	336143.27	0.47	0.11	3
PHR	Loc:Hybrid	2.33	0.69	0.05	3
PHR	Hybrid	3.74	0.69	0.05	3
PHR	Loc	0.05	0.69	0.05	3
PHR	Residual	2.65	0.69	0.05	3
GC	Loc:Hybrid	1.73	0.54	0.63	3
GC	Hybrid	1.48	0.54	0.63	3
GC	Loc	0.06	0.54	0.63	3
GC	Residual	1.96	0.54	0.63	3
PH	Loc:Hybrid	1.96	0.76	0.06	3
PH	Hybrid	9.66	0.76	0.06	3
PH	Loc	4.81	0.76	0.06	3
PH	Residual	2.58	0.76	0.06	3

**Table 2 genes-13-01494-t002:** Prediction performance for each environment and across environments (Global) of **dataset 1 (Indica)** in terms of normalized mean square error (NRMSE) and relative efficiency (RE) under four predictors (G, genotypic information; E + G, environment plus genotypic information; E + G + GE, environment plus genotypic plus genotype by environment interaction information; and G + GE, genotypic plus genotype by environment interaction) under sevenfold cross validation. NRMSE_GBLUP, NRMSE_PLS and NRMSE_RF denote the NRMSE under the **GBLUP**, **PLS** and **random forest** models, respectively. RE_PLS and RE_RF denote the relative efficiency (RE) calculated with the NRMSE of the **PLS** and **random forest** models, respectively. RE was calculated by dividing the prediction performance (with NRMSE) of the **GBLUP** model between the prediction performance of the **PLS** and **random forest** models; that is, the **GBLUP** model was considered the reference model.

Data	Predictor	Env	NRMSE_GBLUP	NRMSE_PLS	NRMSE_RF	RE_PLS	RE_RF
Indica	G	2010	0.892	0.981	0.907	0.909	0.984
Indica	G	2011	1.040	1.079	1.046	0.964	0.994
Indica	G	2012	0.917	1.001	0.966	0.916	0.949
Indica	G	Global	0.880	0.948	0.900	0.928	0.978
Indica	E + G	2010	0.876	0.969	0.853	0.904	1.027
Indica	E + G	2011	0.924	0.961	0.900	0.962	1.027
Indica	E + G	2012	0.839	0.901	0.836	0.931	1.004
Indica	E + G	Global	0.817	0.884	0.810	0.925	1.009
Indica	E + G + GE	2010	0.861	0.959	0.869	0.898	0.991
Indica	E + G + GE	2011	0.901	0.964	0.918	0.934	0.982
Indica	E + G + GE	2012	0.840	0.890	0.849	0.944	0.990
Indica	E + G + GE	Global	0.808	0.880	0.827	0.918	0.976
Indica	G + GE	2010	0.874	0.957	0.877	0.913	0.996
Indica	G + GE	2011	0.910	1.017	0.926	0.895	0.983
Indica	G + GE	2012	0.851	0.914	0.859	0.931	0.990
Indica	G + GE	Global	0.816	0.900	0.833	0.907	0.980

**Table 3 genes-13-01494-t003:** Variance components (variance) and heritability’s estimates for **Japonica** (**dataset 2**) for each trait. CV denotes the coefficient of variation, and Locs denotes the average number of locations.

Trait	Component	Variance	Heritability	CV	Locs
GY	Loc:Hybrid	186065.908	0.29	0.16	3.60
GY	Hybrid	257287.998	0.29	0.16	3.60
GY	Loc	1860782.427	0.29	0.16	3.60
GY	Residual	272836.420	0.29	0.16	3.60
PHR	Loc:Hybrid	0.0001	0.46	0.07	3.60
PHR	Hybrid	0.0004	0.46	0.07	3.60
PHR	Loc	0.0012	0.46	0.07	3.60
PHR	Residual	0.0003	0.46	0.07	3.60
GC	Loc:Hybrid	0.000	0.25	0.82	3.60
GC	Hybrid	0.001	0.25	0.82	3.60
GC	Loc	0.006	0.25	0.82	3.60
GC	Residual	0.001	0.25	0.82	3.60
PH	Loc:Hybrid	0.002	0.62	0.10	3.60
PH	Hybrid	20.528	0.62	0.10	3.60
PH	Loc	35.950	0.62	0.10	3.60
PH	Residual	8.576	0.62	0.10	3.60

**Table 4 genes-13-01494-t004:** Prediction performance for each environment and across environments (global) of **dataset 2 (Japonica)** in terms of normalized mean square error (NRMSE) and relative efficiency (RE) under four predictors (G, genotypic information; E + G, environment plus genotypic information; E + G + GE, environment plus genotypic plus genotype by environment interaction information; and G + GE, genotypic plus genotype by environment interaction), under sevenfold cross validation. NRMSE_GBLUP, NRMSE_PLS and NRMSE_RF denote the NRMSE under the **GBLUP**, **PLS** and **random forest** models, respectively. RE_PLS and RE_RF denote the relative efficiency (RE) calculated with the NRMSE of the **PLS** and **random forest** models, respectively. RE was calculated by dividing the prediction performance (with NRMSE) of the **GBLUP** model between the prediction performance of the **PLS** and **random forest** models; that is, the **GBLUP** model was considered the reference model.

Data	Predictor	Env	NRMSE_GBLUP	NRMSE_PLS	NRMSE_RF	RE_PLS	RE_RF
Japonica	G	2009	2.469	2.511	2.793	0.983	0.884
Japonica	G	2010	2.269	2.263	2.387	1.003	0.951
Japonica	G	2011	1.350	1.349	1.445	1.000	0.934
Japonica	G	2012	2.054	1.943	2.124	1.057	0.967
Japonica	G	2013	1.263	1.348	1.356	0.937	0.932
Japonica	G	Global	0.957	0.983	1.056	0.973	0.906
Japonica	E + G	2009	0.975	1.040	0.998	0.938	0.977
Japonica	E + G	2010	0.927	0.972	0.834	0.954	1.112
Japonica	E + G	2011	0.790	0.914	0.851	0.864	0.928
Japonica	E + G	2012	0.842	0.931	0.918	0.904	0.916
Japonica	E + G	2013	0.778	0.969	0.906	0.803	0.859
Japonica	E + G	Global	0.465	0.564	0.542	0.823	0.857
Japonica	E + G + GE	2009	1.008	1.069	0.985	0.943	1.024
Japonica	E + G + GE	2010	0.819	1.008	0.872	0.812	0.939
Japonica	E + G + GE	2011	0.796	0.924	0.858	0.862	0.928
Japonica	E + G + GE	2012	0.872	0.949	0.945	0.919	0.922
Japonica	E + G + GE	2013	0.782	0.978	0.919	0.800	0.851
Japonica	E + G + GE	Global	0.478	0.578	0.557	0.828	0.859
Japonica	G + GE	2009	1.064	1.581	0.992	0.673	1.072
Japonica	G + GE	2010	0.848	1.794	0.877	0.473	0.968
Japonica	G + GE	2011	0.791	1.032	0.859	0.767	0.921
Japonica	G + GE	2012	0.867	1.283	0.940	0.675	0.922
Japonica	G + GE	2013	0.766	1.120	0.920	0.684	0.833
Japonica	G + GE	Global	0.478	0.720	0.556	0.663	0.859

## Data Availability

The genomic and phenotypic data for the six datasets (datasets 1–6) included in this study are available at https://hdl.handle.net/11529/10548728. This link also includes Appendix A with figures and tables for dataset 3-6, as well as R codes for SKM.

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
