# Peer review of "A Comparison of Three Machine Learning Methods for Multivariate Genomic Prediction Using the Sparse Kernels Method (SKM) Library"

_genes, 2022, doi:10.3390/genes13081494_

Round 1

Reviewer 1 Report

In the manuscript “A comparison between three machine learning methods for multivariate genomic prediction using the Sparse Kernels Methods (SKM) library”, Lopez et al. address the use of multiple models traditionally used for genomic prediction. These models are implemented in a multi-response fashion such that multi-trait genomic prediction may be performed. The area of multi-trait genomic prediction is a growing area of research that is of immense interest with the potential to revolutionize plant breeding. The authors do an excellent job of detailing each of the methods and their corresponding strengths and weaknesses. Data and code are both provided, which is crucial for both reproducibility and ongoing educational purposes. However, with these strengths come several shortcomings that are addressed further below. 

When discussing genomic prediction, it is common and important that the heritability of traits be discussed. If much of the phenotypic variability is not explained by genetics, then one would not expect to accurately predict individual traits, let alone multiple traits. The heritability of traits is not provided here nor are correlations among traits discussed. These aspects should be discussed as they also better inform not only the potentially expected results but may also provide additional reasons for the varied model performance under different predictors. 

While the scale of analyses performed here is commendable, the lack of any substantial discussion concerning the groundnut/wheat analysis results leaves the question of their inclusion wanting. There should be at least some mention of general results/conclusions within the discussion section highlighting the consistency of results across the datasets. 

The SKM library has been previously published. However, there are sections of this manuscript that read more like an introduction to the SKM library. You may very well be promoting your own manuscript (which is fine), but there are likely sections describing the library don’t need inclusion here. That being said, I believe the original SKM manuscript indicates that the MT models require continuous response variables. I don’t recall seeing that point made here, but it seems like that should be included. 

Figures describing the predictive ability of the models represent bar graphs without any discernible measure of predictive variability. This makes sense given the predictive ability is based upon a single held out portion of the data. However, in many cases, model predictive performance is very similar. So similar that one may question whether the differences are very significant. Certainly, any improvement in predictive ability is desirable. But even a description of the cross-validation error would provide some insight into model consistency/variability. 

It would be good if results were able to determine which traits most affected predictive ability within a given multi-trait model. If only some models are able to address this point, then indicating that would be valuable. One might expect again some relationship between heritability and predictive performance. 

While much (nearly all; all?) of the data used here has been previously published, there should be some description of the quality control performed on phenotypic/genotypic data. This is a particularly poor oversight in my opinion. There may also be some discrepancy in the described data. The Indica and Japonica genetic data descriptions both say that there were 16,383 SNPs after QC. This seems highly unlikely especially given that the original data had 92,430 and 44,598 SNPs for Indica and Japonica, respectively. 

Other minor points: 

The first sentence of your abstract is very long... 

Line 37, can we still say GS is “novel”? 

Lines 40-41, you should indicate that phenotype predictions are performed using genotypic data. 

Lines 44-45, GS doesn’t reduce the genotyping technology costs, but it may allow more directed sequencing and thereby reduce the cost of GS. 

Lines 45-47, I think I get what you’re saying here, but it needs to be clarified better. 

Lines 48-88. This whole section uses a single citation (from your book). While that’s not technically wrong, it might be a bit presumptuous to only cite yourself for an entire page. Additional citations are also beneficial for those attempting to better understand the material at hand. 

Line 147, I believe you mean “multi-variate normal” 

Line 184, citation #20 is skipped and Wold is referenced as #20 (it’s 21). 

Line 297, Pandey et al. 2020 [26]. At least I believe that’s what you mean. 

Lines 317-323, some aspects of the description in this paragraph are confusing. Please rephrase/clarify. 

Lines 327-329, the description of the datasets here is sometimes confusing (ex. Dataset 1 – for wheat? Is Indica dataset 1?). Please rephrase/clarify. 

Line 355, “In the case of the Bayesian model (model 1)..” or something like that. 

Line 363, The Bayesian optimization approach could use some more description. This optimization step is strongly dependent on the number of iterations performed, and I don’t think that’s indicated here. 

Lines 406-407, “tunnable” do you mean “tunable”? It doesn’t need to be bold. Though, I realize those are the parameters you are optimizing. 

Line 418, Change to “Use help( 

Line 433, Change to “gs_summaries()” 

Lines 478-484+, your description of the model performance needs a little rephrasing.  

Line 509, Fandom -> “Random”. Unless you have a forest of fans, which is pretty neat as well. 

In your conclusions you state that these findings are only valid for 7-fold CV, but it’s not clear to me what you mean. In what sense are they only valid for 7-fold CV? 

Author Response

RESPONSE TO REVIEWER 1 Comments and Suggestions for Authors

In the manuscript “A comparison between three machine learning methods for multivariate genomic prediction using the Sparse Kernels Methods (SKM) library”, Lopez et al. address the use of multiple models traditionally used for genomic prediction. These models are implemented in a multi-response fashion such that multi-trait genomic prediction may be performed. The area of multi-trait genomic prediction is a growing area of research that is of immense interest with the potential to revolutionize plant breeding. The authors do an excellent job of detailing each of the methods and their corresponding strengths and weaknesses. Data and code are both provided, which is crucial for both reproducibility and ongoing educational purposes. However, with these strengths come several shortcomings that are addressed further below. 

RESPONSE: Many thanks for the time invested in reading and correcting this manuscript. We highly appreciated this extensive, and detailed effort. We have adopted most of the suggestions for improving the quality of the manuscript.

When discussing genomic prediction, it is common and important that the heritability of traits be discussed. If much of the phenotypic variability is not explained by genetics, then one would not expect to accurately predict individual traits, let alone multiple traits. The heritability of traits is not provided here nor are correlations among traits discussed. These aspects should be discussed as they also better inform not only the potentially expected results but may also provide additional reasons for the varied model performance under different predictors. 

RESPONSE: Correction done in the new version of the paper. See lines 483-484 in Table 1 and in lines 600-601 in Table 3.

While the scale of analyses performed here is commendable, the lack of any substantial discussion concerning the groundnut/wheat analysis results leaves the question of their inclusion wanting. There should be at least some mention of general results/conclusions within the discussion section highlighting the consistency of results across the datasets. 

RESPONSE: Correction done in the new version of the paper. See lines 760-763.

The SKM library has been previously published. However, there are sections of this manuscript that read more like an introduction to the SKM library. You may very well be promoting your own manuscript (which is fine), but there are likely sections describing the library don’t need inclusion here. That being said, I believe the original SKM manuscript indicates that the MT models require continuous response variables. I don’t recall seeing that point made here, but it seems like that should be included. 

RESPONSE: We were very careful that those segments that were highlighted in this manuscript were not well highlighted in the paper of the library (Purpose Machine Learning R Library for Sparse Kernels Methods With an Application for Genome-Based Prediction).

Figures describing the predictive ability of the models represent bar graphs without any discernible measure of predictive variability. This makes sense given the predictive ability is based upon a single held out portion of the data. However, in many cases, model predictive performance is very similar. So similar that one may question whether the differences are very significant. Certainly, any improvement in predictive ability is desirable. But even a description of the cross-validation error would provide some insight into model consistency/variability. 

RESPONSE: Yes, we not put the Standard errors to the bars.

It would be good if results were able to determine which traits most affected predictive ability within a given multi-trait model. If only some models are able to address this point, then indicating that would be valuable. One might expect again some relationship between heritability and predictive performance. 

RESPONSE: done see lines 794-798.

While much (nearly all; all?) of the data used here has been previously published, there should be some description of the quality control performed on phenotypic/genotypic data. This is a particularly poor oversight in my opinion. There may also be some discrepancy in the described data. The Indica and Japonica genetic data descriptions both say that there were 16,383 SNPs after QC. This seems highly unlikely especially given that the original data had 92,430 and 44,598 SNPs for Indica and Japonica, respectively. 

RESPONSE: Done. See lines 290-293.

Other minor points: 

The first sentence of your abstract is very long... 

RESPONSE: Correction done in the new version of the paper. See lines 15-16.

Line 37, can we still say GS is “novel”? 

RESPONSE: Yes it was corrected. See line 37.

Lines 40-41, you should indicate that phenotype predictions are performed using genotypic data. 

RESPONSE: Done. See lines 41-42.

Lines 44-45, GS doesn’t reduce the genotyping technology costs, but it may allow more directed sequencing and thereby reduce the cost of GS. 

RESPONSE:Correction done in the new version of the paper. See lines 44-45.

Lines 45-47, I think I get what you’re saying here, but it needs to be clarified better. 

RESPONSE: See lines 47.

Lines 48-88. This whole section uses a single citation (from your book). While that’s not technically wrong, it might be a bit presumptuous to only cite yourself for an entire page. Additional citations are also beneficial for those attempting to better understand the material at hand. 

RESPONSE: Yes you are correct and appreciate it very much your comment. See lines 48-98.

Line 147, I believe you mean “multi-variate normal” 

RESPONSE: It is correct since we are taking about matrix-variate normal distribution, not about the multivariate normal distribution, for this reason it is not required any correction.

Line 184, citation #20 is skipped and Wold is referenced as #20 (it’s 21). 

RESPONSE: See line 187.

Line 297, Pandey et al. 2020 [26]. At least I believe that’s what you mean. 

RESPONSE Correction done in the new version of the paper. See lines 305.

Lines 317-323, some aspects of the description in this paragraph are confusing. Please rephrase/clarify. 

RESPONSE Yes. See lines 324-328.

Lines 327-329, the description of the datasets here is sometimes confusing (ex. Dataset 1 – for wheat? Is Indica dataset 1?). Please rephrase/clarify. 

RESPONSE: Corrected. See lines 333-334.

Line 355, “In the case of the Bayesian model (model 1)..” or something like that. 

RESPONSE; Done. See lines 361.

Line 363, The Bayesian optimization approach could use some more description. This optimization step is strongly dependent on the number of iterations performed, and I don’t think that’s indicated here. 

RESPONSE:Done it.. See lines 370-378.

Lines 406-407, “tunnable” do you mean “tunable”? It doesn’t need to be bold. Though, I realize those are the parameters you are optimizing. 

RESPONSE: Correction done in the new version of the paper. See lines 421.

Line 418, Change to “Use help(“ 

RESPONSE Yes. Thanks. See lines 433.

Line 433, Change to “gs_summaries()” 

RESPONSE: It was changed. See lines 448.

Lines 478-484+, your description of the model performance needs a little rephrasing.  

RESPONSE: Correction done in the new version of the paper. See lines 514-518.

Line 509, Fandom -> “Random”. Unless you have a forest of fans, which is pretty neat as well. 

RESPONSE. Done. Correction done in the new version of the paper. See line 543.

In your conclusions you state that these findings are only valid for 7-fold CV, but it’s not clear to me what you mean. In what sense are they only valid for 7-fold CV? 

RESPONSE: Hope it was clarified. See lines 825-827.

Reviewer 2 Report

The study addresses a relevant topic in plant breeding which is genomic prediction. More specifically, the authors compared the prediction ability of three methods for multivariate genomic prediction. 

I have some suggestions that were included as comments in the attached pdf file. I hope it is useful.

Author Response

RESPONSE REVIWER 2

RESPONSES: The author thank very much the reviewers for the time and effort to read and comment this manuscript. This is highly appreciated and valued by the authors of the article

Does mixed-models be a machine learning method?

 RESPONSE: Yes, Linear Mixed Models are a type of statistical learning method. Clarifications added. See lines 57-58.

Please, give feedback to readers about what is a 'small' data set

 RESPONSE Correction done in the new version of the paper. See lines 68-70.

Only empirical? There is not any study that compared MT models with ST analysis?

RESPONSE: See lines 92-93.

please, give more details about how these covariates were measured

RESPONSE: Done. See lines 287-288.

for minor and major allele? Please, clarify

RESPONSE: See lines 290-293.

This is data set 1, please make the correction.

RESPONSE: Correction done. See line 298.

Data set 4 is this correct?

RESPONSE: Thanks;  this is Data set 1, and it was corrected. See lines 300 and 301.

single nucleotide polymorphism (SNP)

RESPONSE: See lines 312 and 313.

Replace several by six

RESPONSE: Yes and thanks. See line 321.

Replace  single nucleotide polymorphism by only SNPs

RESPONSE: Yes and thanks. See line 324.

SKM is not on CRAN,

RESPONSE: Yes, but can be installed from GitHub. See lines 454-459.

Before this topic, authors could include a comparison of the computational efforts for each method (in terms of computation time and memory usage). Since I could not download the data (only scripts) I have no idea about the method's performance in terms of computational efforts. I believe that readers would also find this information very useful.

RESPONSE:. See lines 470-473.

I could not download the "*.RData" files since they have restricted access.

RESPONSE: All genomic and phenotypic data for all the data sets included in the manuscript together with the Supplementary Materials can be downloaded. The only thing you should do is press the bottom DOWNLOAD and filled out few simple lines of information that are asked by CIMMYT.

Just one suggestion. If the authors could prepare a website like the followin

RESPONSE: thanks, you for the suggestion. CIMMYT as a full public institution freely share data, germplasms, acknowledgements, software, information, act. with anyone in the world. However, through DATAVERSE SYSTEM CIMMYT like to preserve the quality of the share data and decided to have a unified open system. Colleagues interesting in downloading all the data and the software only need to fill out few lines of information – very simple and rapid to do.

Figure 1 and table 1 shows the same results for the NRMSE. Please, keep only one of them (the same with figure 2 and table 2

RESPONSE. Thanks. However, we would like to maintain both figures and tables because figures show the general trends, whereas tables depict small detains.

please, keep using bold for all this indications

RESPONSE:. See yellow parts of GBLUP, Random forest and PLS in lines 475-714.

Would be "supplementary portion"?

RESPONSE: portion or segment.; both seem all right.. See lines 737-738.

This makes sense. How the 18 environmental covariates mentioned in the material and methods were actually used? They were used to produce the environmental relationship matrix mentioned in Line 154?

RESPONSE: Yes. See lines 251-254.

If the E + G or E + G + GE where the better models, it is possible to assume that there are different relationship levels between genotypes across environments according to the envirotyping-based kinships. Therefore, showing an enviromic kernel using the climate covariates would enrich the discussion and provide insights on how to improve the genomic prediction using climate information.

RESPONSE: See lines 753-757.

These papers by Costa-Neto et al. can be useful Costa-Neto, G., R. Fritsche-Neto, and J. Crossa. 2021. Nonlinear kernels, dominance, and envirotyping data increase the accuracy of genome-based prediction in multi-environment trials. Heredity 126(1): 92–106. doi: 10.1038/s41437-020-00353-1. Costa-Neto, G.,

  1. Galli, H.F. Carvalho, J. Crossa, and R. Fritsche-Neto. 2021. EnvRtype: a software to interplay enviromics and quantitative genomics in agriculture. G3 Genes|Genomes|Genetics 11(4). doi: 10.1093/G3JOURNAL/JKAB040

RESPONSE: Yes, fully agree. Many thanks. See line 756.

Since they differ regarding the degree of correlation between traits and non-linear patterns, readers would like to know how are the opinion of the authors about the better method to be used. In practice, what has more impact: correlation between traits or non-linear patterns?

RESPONSE: Yes agree. See lines 790-792.

Additionally, I know that this was not a goal of the study, but a comparison to single-trait prediction models would give a clear notion of the efficiency of MT-models implemented here. Could the authors provide such a comparison?

RESPONSE: Yes indeed but we did not include it to avoid distraction from the main topic. The article is already very extensive

Round 2

Reviewer 1 Report

Only minor comments at this point, 

I still think you should add standard error to the bar plots. Also, data like this are better represented as scatter plots than bar plots. https://github.com/cxli233/FriendsDontLetFriends#1-friends-dont-let-friends-make-bar-plots-for-means-separation (or https://journals.plos.org/plosbiology/article?id=10.1371/journal.pbio.1002128) 

Line 44: “catalyzed by a significant...” 

Line 96: Parsimonious 

Line 423: Also “tunable” not “tunnelable” 

Line 738: Supplementary 

You use both “nonlinear” and “non-linear” at times. Be consistent. 

Author Response

RESPONSE TO REVIEWER 2

Open Review

Only minor comments at this point, 

I still think you should add standard error to the bar plots. Also, data like this are better represented as scatter plots than bar plots. https://github.com/cxli233/FriendsDontLetFriends#1-friends-dont-let-friends-make-bar-plots-for-means separation (or https://journals.plos.org/plosbiology/article?id=10.1371/journal.pbio.1002128) 

RESPONSE: Yes, indeed bars were added to all figures in the paper as well as those in the SUPPLEMENTARY MATERIALS

Line 44: “catalyzed by a significant...” 

RESPONSE Correction done

Line 96: Parsimonious 

RESPONSE Correction done

Line 423: Also “tunable” not “tunnelable” 

RESPONSE Correction done

Line 738: Supplementary 

RESPONSE Correction done

You use both “nonlinear” and “non-linear” at times. Be consistent. 

RESPONSE Correction done

Submission Date

13 July 2022

Date of this review

08 Aug 2022 16:55:48
